# Molecular Characterization of Some *Bacillus* Species from Vegetables and Evaluation of Their Antimicrobial and Antibiotic Potency

**DOI:** 10.3390/molecules28073210

**Published:** 2023-04-04

**Authors:** Moldir Koilybayeva, Zhanserik Shynykul, Gulbaram Ustenova, Symbat Abzaliyeva, Mereke Alimzhanova, Akerke Amirkhanova, Aknur Turgumbayeva, Kamilya Mustafina, Gulnur Yeleken, Karlygash Raganina, Elmira Kapsalyamova

**Affiliations:** 1School of Pharmacy, S.D. Asfendiyarov Kazakh National Medical University, Tole-bi 94, Almaty 050012, Kazakhstan; 2Higher School of Medicine, Al-Farabi Kazakh National University, Tole-bi 96, Almaty 050040, Kazakhstan; 3Center of Physical Chemical Methods of Research and Analysis, Al-Farabi Kazakh National University, Tole-bi 96, Almaty 050012, Kazakhstan; 4School of Medicine, S.D. Asfendiyarov Kazakh National Medical University, Tole-bi 94, Almaty 050012, Kazakhstan

**Keywords:** antimicrobial activity, antibiotic substances, *Bacillus subtilis*, *Bacillus subtilis*, Khozestan2, bacterial isolates, GC–MS

## Abstract

Numerous natural habitats, such as soil, air, fermented foods, and human stomachs, are home to different *Bacillus* strains. Some *Bacillus* strains have a distinctive predominance and are widely recognized among other microbial communities, as a result of their varied habitation and physiologically active metabolites. The present study collected vegetable products (potato, carrot, and tomato) from local markets in Almaty, Kazakhstan. The bacterial isolates were identified using biochemical and phylogenetic analyses after culturing. Our phylogenetic analysis revealed three Gram-positive bacterial isolates BSS11, BSS17, and BSS19 showing 99% nucleotide sequence similarities with *Bacillus subtilis* O-3, *Bacillus subtilis* Md1-42, and *Bacillus subtilis* Khozestan2. The crude extract was prepared from bacterial isolates to assess the antibiotic resistance potency and the antimicrobial potential against various targeted multidrug-resistant strains, including *Staphylococcus aureus*, *Staphylococcus epidermidis*, *Streptococcus group B*, *Streptococcus mutans*, *Candida albicans*, *Candida krusei*, *Pseudomonas aeruginosa*, *Shigella sonnei*, *Klebsiella pneumoniae*, *Salmonella enteritidis*, *Klebsiella aerogenes*, *Enterococcus hirae*, *Escherichia coli*, *Serratia marcescens*, and *Proteus vulgaris*. This study found that the species that were identified have the ability to produce antibiotic chemicals. Additionally, the GC–MS analysis of three bacterial extracts revealed the presence of many antibiotic substances including phenol, benzoic acid, 1,2-benzenedicarboxylic acid and bis(2-methylpropyl), methoxyphenyl-oxime, and benzaldehyde. This work sheds light on the potential of *Bacillus* to be employed as an antimicrobial agent to target different multidrug-resistant bacterial strains. The results indicate that market vegetables may be a useful source of strains displaying a range of advantageous characteristics that can be used in the creation of biological antibiotics.

## 1. Introduction

The prevalence of antibiotic resistance in bacterial strains poses a severe threat to public health and calls for urgent research into novel antibiotics or antimicrobial chemicals [1]. Numerous studies on the development of novel antibiotics from various microbe and plant strains have been published over the past few decades [2,3,4,5]. Antibiotics are chemicals that bacteria create and use in their natural environments for protection from the invasion of other bacterial species. In addition to serving as a kind of protection, these antibiotics are essential signaling molecules that allow the cells of the bacterial population to communicate with one another [6,7,8]. Their importance is enhanced when a bacterium is considered a probiotic which is administered in sufficient amounts and provides health benefits to the host. Cases of affirmed health profits of probiotics may comprise the support of immune health, gastrointestinal health, and so on. More precisely, the probiotic *Bacillus subtilis* has clinically demonstrated its effectiveness in dietary protein digestion [9]. Thus, it is impossible not to notice the growing number of benefits associated with the use of bacteria in various fields of activity. However, it is unfair not to consider the side effects of antibiotic compounds synthesized by bacteria. It was found that antibiotics influence food deterioration. Thus, spoilage of food is caused by a change in the chemical or physical properties of food caused by antibiotic-like substances of foodborne pathogens [10].

From the historical evidence, it is assumed that natural products are essential for the discovery and advancement of antibiotics [11]. It is imperative to investigate innovative antimicrobial substances with a high potential to eradicate or control a variety of microorganisms. Hence, one of the fundamental pillars of modern medicine is the antibiotic. Nevertheless, it is regrettable that some pathogenic strains render commonly-used antibiotics ineffective, and there is a requirement for novel antibiotics to take their place [12,13]. Microorganisms that may create bioactive secondary metabolites have unique structural features and biological activities. These bioactive compounds are produced by a few types of microflora found in vegetables and are employed as antibiotics. Several other notable types of investigation have also been reported to identify bacteria from vegetables with new antimicrobial agents [14,15]. Multidrug-resistant infections are more hazardous than infections caused by bacterial pathogens that are not resistant to multiple drugs because public health practitioners have recently had tremendous difficulty in treating these organisms.

Particularly, the prevalence of resistance developed in bacterial pathogens functions as a secondary infection in a number of life-threatening disorders, such as cancer, surgical procedures, transplantation, etc., and affects the effectiveness of contemporary treatments in treating these conditions [16,17]. It is evident that there are very few therapeutic drugs available to successfully treat these infections given the rapid evolution of multidrug-resistant strains due to the availability of relatively few effective treatments [18,19]. In order to evaluate the antibacterial properties against the majority of common human diseases, this study concentrated on extracting possible bacterial species from vegetable sources (potato, carrot, and tomato) and creating a crude extract from isolated bacterial strains. The main goal of this study is to provide insights into bacterial isolate composition and to provide a connection with the antimicrobial activity of investigating bacterial strains. The bioactive components in the crude extract were also identified using GC–MS. This research will aid in the creation of new drugs to combat multidrug-resistant bacterial strains.

## 2. Results

### 2.1. Isolation and Identification

A total of *n* = 19 bacteria strains were isolated and identified based on colonial morphology, microscopy, biochemical characteristics, and sugar fermentation. Among all, Gram-positive, rod-shaped, mycelial, and spore-forming bacterial strains were selected for further confirmatory tests. The molecular analysis further validated the bacterial strains (BSS11, BSS17, and BSS19) as *Bacillus subtilis* O-3, *Bacillus subtilis* Md1-42, and *Bacillus subtilis* Khozestan2.

### 2.2. Morphological Characterization

The morphology of each colony from the different bacterial isolates showed regular, irregular, slightly raised, flat, white, and cream-colored colonies. A motility test determined that the bacterial isolates were motile and possessed terminal and subterminal spores (Table 1).

### 2.3. Antimicrobial Activity Assessment

All tested isolates of BSS11, BSS17, and BSS19 showed antagonistic activity against most bacterial pathogens, such as *Klebsiella aerogenes* ATCC 13048, *Staphylococcus aureus* ATCC 29213, *Staphylococcus epidermidis* ATCC 12228, *Candida krusei* ATCC 14243, and *Candida albicans* ATCC 2091, and had less activity against other pathogens, such as *Proteus vulgaris* ATCC 6380, *Klebsiella pneumoniae* ATCC 13883, *Shigella sonnei* ATCC 25931, and *Salmonella enterica* ATCC 35664 (Figure 1).

The nine extracts showed antibacterial activity against all the bacterial pathogens except *Shigella sonnei* ATCC 25931, *Klebsiella pneumonia* ATCC 13883, *Salmonella enterica* ATCC 35664, and *Enterococcus hirae* ATCC 10541 (Table 2). Additionally, three extracts such as EAE (C), EAE (BC), and EAE (SC) from *Bacillus subtilis* Khozestan2 (BSS19) did not also demonstrate antibacterial activity. The nine extracts showed antibacterial activity against all the bacterial pathogens except *Shigella sonnei* ATCC 25931, *Klebsiella pneumonia* ATCC 13883, *Salmonella enterica* ATCC 35664, and *Enterococcus hirae* ATCC 10541. Additionally, three extracts such as EAE (C), EAE (BC), and EAE (SC) from *Bacillus subtilis* Khozestan2 (BSS19) have not demonstrated any antibacterial activity, too. The EAE (SC) preparation of *Bacillus subtilis* O-3 (BSS11), showed a better zone of inhibition for *Staphylococcus epidermidis* ATCC 12228 (25 ± 1.20 mm), *Streptococcus* group B (19 ± 1.20 mm), *Candida krusei* ATCC 14243 (29 ± 2.35 mm), *Klebsiella aerogenes* ATCC 13048 (17 ± 1.82 mm), and *Proteus Vulgaris* ATCC 6380 (18 ± 1.34 mm) when compared with the other pathogens, while the EAE (SC) preparation of *Bacillus subtilis* Md1-42 (BSS17) was effective against the pathogens *Staphylococcus aureus* ATCC 29213 (30 ± 2.50 mm) and *Serratia marcescens* ATCC 13880 (18 ± 1.64 mm). Additionally, the EAE (SC) preparation of *Bacillus subtilis* Khozestan2 (BSS19) was effective against the pathogens *Streptococcus* mutans ATCC 25175 (20 ± 1.32 mm), *Candida albicans* ATCC 2091 (40 ± 1.22 mm), and *Pseudomonas aeruginosa* ATCC 9027 (22 ± 1.81 mm). It was found assumingly the same potency meaning of the EAE (SC) preparations of *Bacillus subtilis* Md1-42 (BSS17) and *Bacillus subtilis* Khozestan2 (BSS19) against the pathogen *Escherichia coli* ATCC 25922.

### 2.4. Antibiotic Susceptibility Profile of the Isolates

The study of the antibiogram revealed that all three tested *Bacillus* subtilis subspecies were resistant to all antibiotics except for bacitromycin (B, 10), polymyxin (PB, 300), and cloxacillin (CX, 5) (Table 3). BSS11, BSS17, and BSS17 strains showed the highest vulnerability to gentamicin (CN, 120) with 40 ± 0.28 mm, 40 ± 0.28 mm, and 38 ± 0.28 sensitivity diameters, respectively, while strain BSS17 showed the lowest sensitivity to carbenicillin (10 ± 0.28) and to amoxycillin (12 ± 0.29) with high significant differences (*p* < 0.0001).

### 2.5. Analysis of the Isolates Using GC–MS

The crude extracts from several *Bacillus* bacterium species contained a number of chemicals, according to the results of the GC–MS analysis. Table 4, Table 5 and Table 6 explain the most significant and abundant components found in the crude extracts that were subjected to the GC–MS analysis, as well as information about where the chemicals found in this study had previously been identified. These substances exhibited similarities to natural products of bacterial and plant origin. According to the study of the GC–MS data, the majority derived from volatile substances, such as alkaloids, esters, ethers, and phenolic chemicals.

The GC–MS analysis of the ethyl acetate extracts of the three isolates detected a total of 106 compounds. Based on the analysis of bacterial isolates, BSS11 and BSS17 were found to share a similar composition of volatile organic components, while bacterial isolate BSS19 was found with fewer quantities of them. For isolate BSS11, the solvent with metabolites was ethyl acetate with 36 compounds (Table 4). Phenol, benzoic acid, phenol, 2,4-bis(1,1-dimethylethyl), 1,2-benzenedicarboxylic acid, methoxyphenyl-oxime, and benzaldehyde (Figure 2) were identified in the BSS11 extract with important concentrations of 4.246424%, 0.94121%, 1.242429%, 2.287847%, 2.008104%, and 7.155098%, respectively. In the extract, the major compounds were acetamide at 11.58042% and 2-butanone at 9.68622%, while minor compounds were fatty acids and their derivatives. In strain BSS17, ethyl acetate extraction showed the presence of 39 compounds (Table 5) in comparison to 32 compounds arising out with the same extraction strain BSS19 (Table 6). GC–MS analysis for two bacterial (BSS17 and BSS19) analyses also confirmed the presence of the same volatile organic compounds but with fewer amounts compared with BSS11. The BSS19 ethyl acetate extract did not reveal the presence of some fatty acids (octanoic acid, nonanoic acid, hexadecanoic acid, octadecanoic acid, oleic acid, and 9,12-octadecanoic acid (z,z)), although they are present in the extracts of BSS11 and BSS17.

### 2.6. Molecular Characterization

From various samples, three bacterial isolates with enhanced antibacterial activity were discovered. Phylogenetic analysis of the 16S rRNA gene sequences revealed that all three candidate bacterial isolates, BSS11, BSS17, and BSS19, belong to three different *Bacillus* species, respectively (Figure 3), as they group together in the evolutionary tree with the aforementioned bacterial species.

*Bacillus subtilis* O-3, *Bacillus subtilis* Md1-42, and *Bacillus subtilis* Khozestan2 were identified as having the highest hit sequence similarity for these bacterial isolates (Table 7). High bootstrap values were obtained following a phylogenetic analysis and tree topology both served to confirm the presumably described taxonomy.

## 3. Discussion

Extreme microbial diversity, abundance, and structure are also correlated with a wide range of metabolic processes, which generate a large number of metabolites with a variety of functions, including antimicrobial, anti-parasitic, anti-cancerous, and anti-pesticidal functions. The current study sought to examine the potential for specific vegetable microbial populations to exhibit antibacterial activities. Nineteen distinct bacterial isolates were discovered, as a result of a number of isolation stages, the identification of different general purposes with the selection of bacterial growth media, and biochemical studies. In recent years, a rise in the likelihood of discovering new antibiotics to combat or control untreated infectious diseases has been seen, thanks to a number of microorganisms that can produce antibiotics when grown in proper cultures [20,21]. Indeed, the development of resistance genes in bacteria through the use of mobile genetic elements or their inherent characteristics (natural phenotypic traits) are the two main causes of their antibiotic resistance [22]. All three *Bacillus* spp. strains were susceptible to almost all antibiotics, except for bacitracin, polymyxin, and cloxacillin, for which they were all resistant (Table 3). Similar findings on the susceptibility of various antibiotic-resistant *Bacillus* species were observed [23,24,25,26]. Our results also agree with those on the resistance of *Bacillus* strains to bacitracin, published by Adimpong et al. (2012) [27]. According to Adimpong et al. (2012) and Compaoré et al. (2013) [28], the resistance of specific *Bacillus* strains to particular antibiotics may be inherent or acquired and associated with the existence of resistance genes implicated in the production of resistance enzymes to these antibiotics. Conversely, the probability of passing on resistance genes to other dangerous bacteria is lower because of a natural resistance, rather than an acquired resistance. Since resistant bacteria can spread from the food chain to humans, antibiotic resistance has really become a serious global concern [29]. Although isolated *Bacillus* strains do not seem to harbor antibiotic resistance genes that can be passed on to dangerous germs, it is possible that they do not respond to a wide variety of antibiotics. Further research into these strains as prospective probiotic starter cultures could improve and maximize the production of high-quality, medicinal, and functional or health-promoting substances.

The antibacterial characteristic plays a crucial role in therapeutic activities. In the present study, the perpendicular streak method was used to determine the antibacterial properties of the bacterial isolates (BSS11, BSS17, and BSS19) against the selected human bacterial pathogens. This approach is regarded as a first-pass qualitative screening technique for the antimicrobial activity. The research demonstrated the strongest antagonistic action against human pathogen and as a result all tested isolates of BSS11, BSS17, and BSS19 showed antagonistic activity against most bacterial pathogens, such as *Klebsiella aerogenes* ATCC 13048, *Staphylococcus aureus* ATCC 29213, *Staphylococcus epidermidis* ATCC 12228, *Candida krusei* ATCC 14243, and *Candida albicans* ATCC 2091.

Our findings demonstrated that the growth of multidrug-resistant bacterial strains is inhibited by *Bacillus subtilis* O-3, *Bacillus subtilis* Md1-42, and *Bacillus subtilis* Khozestan2, which has previously been published [30,31]. Previous research found that the pH of the growing medium or the generation of volatile chemicals are what cause *Bacillus* to have an inhibitory impact. *Bacillus* is known to produce polypeptide antibiotic substances, such as bacitracin, polymyxin, gramicidin S., and tyrothricin, according to a number of other investigations. These substances work well against a variety of bacteria, including Gram-positive and Gram-negative bacteria [32].

GC-MS made it possible to detect markers in the studied samples of biological material—components of a microbial cell and its metabolites (fatty acids, aldehydes, phenolic compounds). Additionally, using GC-MS was beneficial in the case of both endogenous and exogenous microflora, without preliminary isolation of a pure culture of microorganisms, which is especially important when considering the difficulties in cultivating anaerobes. The distinctive advantages of the method were the speed of analysis and the ability to quantify the content of the marker. According to the GC–MS analysis, the *Bacillus* species produce a variety of antifungal chemicals. The synthesis of antibiotic substances by *Bacillus subtilis* O-3, *Bacillus subtilis* Md1-42, and *Bacillus subtilis* Khozestan2 strains was determined by the GC–MS analysis of their crude metabolites. The most important antifungal compounds detected from all three strains were phenol, benzoic acid, phenol, 2,4-bis(1,1-dimethylethyl), 1,2-benzenedicarboxylic acid, and bis(2-methylpropyl) (Table 4, Table 5 and Table 6 and Figure 2). Additionally, the GC–MS analysis showed that all three bacterial isolates contained methoxyphenyl-oxime and benzaldehyde. Previously, methoxyphenyl-oxime was reported as a true specific antibacterial agent that controlled some bacteria [33]. Another study proved that benzaldehyde generated by *Photorhabdus temperata* has insecticidal, great antibacterial, and antioxidant properties [34]. Two bacterial isolates of *Bacillus subtilis* O-3 and *Bacillus subtilis* Md1-42 share common fatty acids, such as octanoic acid, nonanoic acid, hexadecanoic acid, octadecanoic acid, oleic acid, and 9,12-octadecanoic acid (z,z). It is well known that fatty acids and their derivatives have powerful antibacterial and antifungal activities [35]. Because of their great biodegradability, low toxicity, and strong resistance to extremes in pH, salinity, and temperature, they are more environmentally friendly. As food additives, they are accepted. Antifungal fatty acids are less likely to make pathogenic fungi resistant to them [36]. Most of the chemicals identified from three different *Bacillus* species were derived from volatile compounds, such as esters, alkaloids, ethers, and phenolics, and shared structural similarities with natural products of bacterial and plant origin. Many volatile organic compounds are major constituents of the bacterial strain and have been shown to have properties against phytopathogens [37,38]. Our results prove that *Bacillus* spp. share common volatile compounds and complements previous findings related to the study of chemical composition of bacterial strains using GC–MS [39,40,41,42]. Moreover, bacterial strains of *Bacillus subtilis* O-3 and *Bacillus subtilis* Md1-42 containing mentioned above volatile organic compounds (phenol, benzoic acid, phenol, 2,4-bis(1,1-dimethylethyl), 1,2-benzenedicarboxylic acid, methoxyphenyl-oxime, and benzaldehyde) and fatty acid derivatives (octanoic acid, nonanoic acid, hexadecanoic acid, octadecanoic acid, oleic acid, and 9,12-octadecanoic acid (z,z)) showed high anti-microbial activity on different pathogens. However, the bacterial strain *Bacillus subtilis* Khozestan2 demonstrated low antimicrobial potency, and it may be caused by due to the absenteeism of fatty acid derivatives and comparingly low concentrations of volatile organic compounds (phenol (1.456046%), benzoic acid (1.3117335%), phenol, 2,4-bis(1,1-dimethylethyl) (1.085735%), 1,2-benzenedicarboxylic acid (0.986222%), methoxyphenyl-oxime (1.54511%), and benzaldehyde (3.066211%)).

There are many purposes for bacterial extract investigation. For instance, *Bacillus* spp. produce a variety of compounds involved in the biocontrol of plant pathogens and the promotion of plant growth, which makes them potential candidates for most agricultural and biotechnological applications. Moreover, the *Bacillus* strains as a form of probiotics are not generally pathogenic to mammals and appear to have significant potential for clinical use. Nevertheless, *Bacillus* probiotics may also generate toxins and biogenic amines; consequently, their safety is a concern.

A molecular investigation revealed the taxonomy of three different isolated species belonging to three *Bacillus* spp., such as *Bacillus subtilis* O-3, *Bacillus subtilis* Md1-42, and *Bacillus subtilis* Khozestan2. It was tentatively determined that the three most viable candidates of bacterial isolates BSS11, BSS17, and BSS19 belong to *Bacillus subtilis* O-3 (99%), *Bacillus subtilis* Md1-42 (99%), and *Bacillus subtilis* Khozestan2 (99%), respectively, based on a phylogenetic analysis and top hit sequence similarity results, which were supported by a high bootstrap value. In the future, the microbial screening and the isolation of active metabolites against multidrug-resistant strains could be carried out more easily by the identification of the three separate bacterial strains and their antibacterial activity.

## 4. Materials and Methods

### 4.1. Isolation of Potential Strains of the Genus Bacillus spp.

*Bacillus* spp. strains have been isolated from different vegetable sources (potato, carrot, and tomato). A 15 g vegetable sample was homogenized in 100 mL of 0.85% NaCl by shaking at 150 rpm for 15 min. Then, the sample was diluted step-wise and incubated in a water bath for 10 min at 90 °C. The sample was cooled to room temperature and then 0.1 mL samples were loaded onto nutrient agar/meat peptone agar (NA/MPA) plates, which is a nutrient medium for the cultivation of non-fastidious microorganisms. NA/MPA plates were composed of gelatin peptone (5 g/L), bacteriological agar (15 g/L), and meat extract (3 g/L). The plates were incubated for 48 h at 37 °C. The isolated pure strains were refrigerated at −20 °C in nutrient broth (NB) media supplemented with 20% (*v*/*v*) glycerin. Then, the fresh culture was subjected to morphological identification and the slightly raised, flat, white, and cream-colored colonies were selected for further identification. Strain isolates in NB media were useful in further studies, particularly in the preparation of ethyl acetate extract for subjecting GC–MS analysis.

### 4.2. Screening for Antagonistic Activity of Isolated Bacteria against Potent Bacterial Pathogens

A preliminary antibacterial analysis of the isolates was conducted on Mueller–Hinton agar (MHA) plates using the perpendicular streak method against powerful human pathogens. *Staphylococcus aureus* ATCC 29213, *Staphylococcus epidermidis* ATCC 12228, *Streptococcus group B*, *Streptococcus mutans* ATCC 25175, *Candida albicans* ATCC 2091, *Candida krusei* ATCC 14243, *Pseudomonas aeruginosa* ATCC 9027, *Shigella sonnei* ATCC 25931, *Klebsiella pneumoniae* ATCC 13883, *Salmonella enterica* ATCC 35664, *Klebsiella aerogenes* ATCC 13048, *Enterococcus hirae* ATCC 10541, *Escherichia coli* ATCC 25922, *Serratia marcescens* ATCC 13880, and *Proteus vulgaris* ATCC 6380 were employed in this investigation as bacterial pathogens. According to the widely-used method of perpendicular streaks, an exponential culture of the studied antagonist strains was streaked on the surface of an agar medium and incubated at 30 ± 4 °C for 24 h [43]. Then, an exponential culture of the test strain was inoculated perpendicularly from the edge of the cup to the stroke of the grown culture of the antagonist with a stroke by slightly touching the stroke of the antagonist strain. The plate was again incubated under conditions favorable for the growth of the test culture.

The cellular preparations were held relying on the procedure elaborated by Beiranvand et al. (2017) [44]. The chosen isolates underwent 14 days of incubation at 30 ± 4 °C while being cultivated in a nutritional broth. Then, three different techniques were used to extract the cultures. The cultures were used further in preparation of the ethyl acetate extract of culture EAE (C), the ethyl acetate extract of boiled and cooled culture EAE (BC), and the ethyl acetate extract of the sonicated culture EAE (SC). For the creation of the three different extracts, the culture broth was divided into three equal portions. In a separate flask, one part of the culture broth was combined with an equivalent amount of ethyl acetate to create EAE (C). Once separated, the ethyl acetate extract was centrifuged at 8000 rpm for 10 min. The ethyl acetate supernatant was poured into a spotless flask and heated to 50 °C for drying. Two (2) ml of DMSO were used to dissolve the dry extract. Another portion of the media was incubated in boiling water for 5 min, and then cooled for 5 min, to prepare the EAE (BC). This mixture was then diluted 1:1 with ethyl acetate. The samples were then processed using the first technique. The culture was sonicated for three minutes at 130 W to prepare EAE (SC), and extraction was then carried out as instructed in the first procedure. Three extracts were examined for their antibacterial efficacy using the well diffusion method against pathogenic bacteria, including *Staphylococcus aureus* ATCC 29213, *Staphylococcus epidermidis* ATCC 12228, *Streptococcus group B*, *Streptococcus mutans* ATCC 25175, *Candida albicans* ATCC 2091, *Candida krusei* ATCC 14243, *Pseudomonas aeruginosa* ATCC 9027, *Shigella sonnei* ATCC 25931, *Klebsiella pneumoniae* ATCC 13883, *Salmonella enterica* ATCC 35664, *Klebsiella aerogenes* ATCC 13048, *Enterococcus hirae* ATCC 10541, *Escherichia coli* ATCC 25922, *Serratia marcescens* ATCC 13880, and *Proteus vulgaris* ATCC 6380. Sterile saline (0.85% NaCl) with an optical density of 0.5 McFarland standard scale (5 × 106 CFU/mL (CFU—Colony Forming Units/mL) for yeasts and 1.5 × 108 CFU/mL for bacteria) was used to prepare microbial suspensions. Each bacterial pathogen’s zone of inhibition was evaluated and control wells containing 20 μL of streptomycin (1 mg/mL) were used.

### 4.3. Antibiotic Susceptibility of the Bacillus Strains

Using the disk diffusion method, the antibiotic susceptibility of isolated *Bacillus* strains (BSS11, BSS17, and BSS19) was assessed in accordance with the guidelines of the European Committee on Antimicrobial Susceptibility Testing (EUCAST, 2019). *Bacillus* strains were spread-plated using sterile beads on Mueller–Hinton (MH) agar using an aliquot of 1 mL each, at a concentration of 106 CFU/mL (0.5 McFarland, Hi-media, India). The plates were then left to dry for an hour. Then, antibiotic disks were inserted into the agar plates containing an inoculated *Bacillus* strain.

The widths of the inhibition zones surrounding the antibiotic disks were measured using an electronic digital vernier caliper micrometer measuring tool caliber digital ruler (ZHHRHC LCD) following a 24 h incubation period at 37 °C (Hardened, China). This made it possible to identify the strain’s antibiotic susceptibility (S), intermediate resistance (I), or resistance (R) according to the CLSI guidelines (2012) [44,45]. Eighteen antibiotic disks contained a sample each of penicillin G (PEN, 10), ampicillin (AMP, 10), amoxycillin (AMOX, 30), amoxycillin-clavulanic acid (AMC, 30), carbenicillin (CAR, 100), cloxacillin (CX, 5), erythromycin (ERO, 15), azithromycin (AZM, 15), cefepime (FEP, 30), cefepime/clavulanic acid FEC-40, cephalatin (KF, 30), cefotaxime (CTX, 30), gentamicin (CN, 120), streptomycin (STR, 10), tobramycin (TOB, 10), tetracycline (TET, 30), polymyxin (PB, 300), and bacitromycin (B, 10).

### 4.4. Gas Chromatography–Mass Spectrum Analysis of the Metabolites

The volatile compounds extraction for each bacterial strain (*Bacillus subtilis* O-3, *Bacillus subtilis* Md1-42, and *Bacillus subtilis* Khozestan2) was done separately two times from 50 mL of the culture broth with 25 mL ethyl acetate (Sigma-Aldrich, Germany) for 30 min and two extracts were combined. Thereafter, the extract with a volume of 1.5 mL was taken into plastic vials with a volume of 2 mL and placed onto the autosampler tray for analysis using GC–MS. Thermo Scientific GC Focus Series DSQ was used to perform a GC–MS analysis on bacterial secondary metabolites. A steady flow of 1 mL of helium gas per minute was employed as the carrier gas, and an infection volume of 1 L was used. The injector and hot oven were kept at 250 °C and 110 °C, respectively, with the temperature increasing by 10 °C per minute up to 200 °C, 5 °C per minute up to 280 °C, and shutting down after 9 min at a temperature of 280 °C. The GC column was used to elute peaks of various chemicals, and the retention times of these peaks were noted. The database was searched for compounds with similar molecular masses and retention times after the data were matched with the compounds’ mass spectra. The bioactivities of previously investigated natural substances were also investigated, and the current study found a comparable correlation between the bioactivities of the bacterial extracts and their constituent parts.

### 4.5. Molecular Characterization of the Bacterial Isolates

Based on 16S rRNA conserved gene sequences and universal bacterial primers, isolated bacterial strains were molecularly characterized. The targeted gene sequence was amplified using the standard PCR procedure, and the final product was run through 1% gel electrophoresis to examine the size of the amplified fragments. The amplified samples and the relevant sequencing fragments were sent for sequencing, and MEGA software was used to phylogenetically analyze the nucleotide sequences that were recovered (MEGA-11). Using GenBank NCBI’s BLAST search, the bacterial isolates were further verified and classified at the species level (National Center for Biotechnology Information). With the accession numbers GQ870259, MF581448, and MH036316, 16S rRNA gene sequences for these probiotic strains were uploaded to the GenBank database (www.ncbi.nlm.nih.gov/projects/genome/clone/, accessed on 9 December 2022).

### 4.6. Statistical Analyses

The XLSAT software version 2016.02.27444 was used to conduct the analysis of variance (one-factor ANOVA) at the significance level (α = 0.05). The Newman–Keuls test was used to rank the means when there was a significant difference between the studied parameters.

## 5. Conclusions

The growth of multidrug-resistant bacterial strains could be inhibited by vegetable bacterial isolates from three different species of *Bacillus*, according to the current study. When examined using well diffusion and the perpendicular streak method, the crude extracts from three isolated bacterial strains were effective against bacterial strains. The potent isolates BSS11, BSS17, and BSS19 with broad-spectrum antibacterial activities were identified through this screening. The metabolic diversity within isolates was highlighted by a comparative GC–MS analysis, despite the fact that they are all members of the same *Bacillus* subspecies. In particular, it is evident in the case of *Bacillus subtilis* O-3 and *Bacillus subtilis* Md1-42. This study discovered a number of volatile inhibitory substances, including esters, phenolics, and ethers that may be involved in antimicrobial activity. It was found how they differ in chemical composition and how it may influence antimicrobial activity and antibiotic potency. For decades, it has been known that strains of the B. subtilis group are capable of producing a variety of secondary metabolites that mediate their antimicrobial characteristics. Along with volatile organic compounds, the presence of bacteriocins, polyketides, peptides, etc., were known. Hence, it can be concluded that the discovered organic volatile substances enhance the antimicrobial properties of *Bacillus* spp. together with the above substances. Bacterial strains *Bacillus subtilis* O-3, *Bacillus subtilis* Md1-42, and *Bacillus subtilis* Khozestan2 exhibit a tremendous metabolic capacity and adaptive biochemistry that could be employed in a variety of commercial and biotechnological activities by generating a wide range of bioactive chemical substances. Additionally, bacterial extracts, including chemicals, could be employed as antimicrobial agents to target different multidrug-resistant bacterial strains. It is anticipated that a thorough investigation of a similar kind could investigate new microbiological possibilities with undiscovered substances or metabolites that have a strong antibacterial potential. As a result, it might be a viable strategy for lowering the burden and danger posed by bacterial strains that are resistant to many drugs.

## Figures and Tables

**Figure 1 molecules-28-03210-f001:**
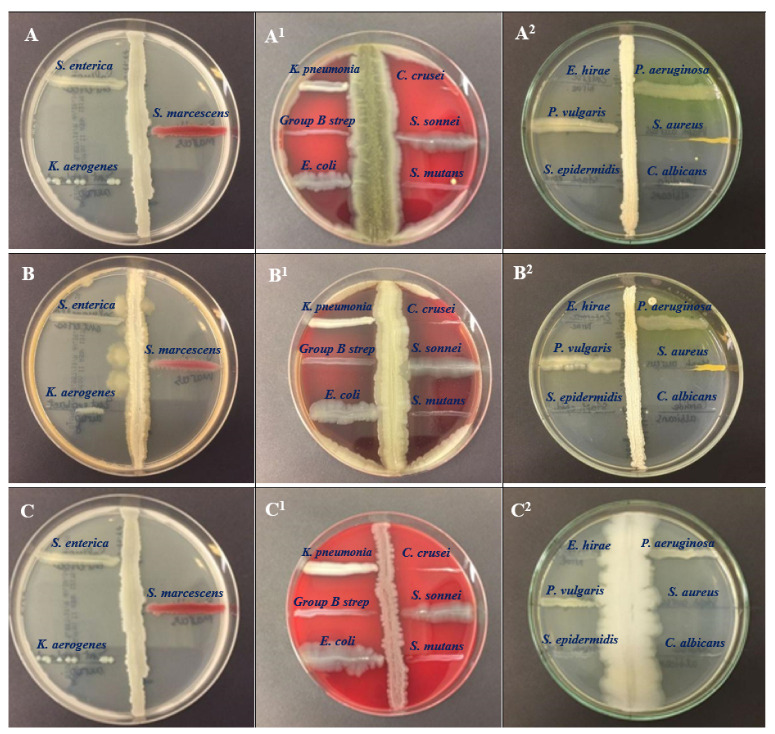
Antagonistic activity of the bacteria of the genus *Bacillus* against pathogens. Antagonistic efficacy of all three isolates was examined against pathogenic bacteria, such as *Salmonella enterica* ATCC 35664, *Serratia marcescens* ATCC 13880, *Klebsiella aerogenes* ATCC 13048, *Candida krusei* ATCC 14243, *Shigella sonnei* ATCC 25931, *Streptococcus mutans* ATCC 25175, *Klebseiella pneumoniae* ATCC 13883, *Group B Streptococcus*, *Escherichia coli* ATCC 25922, *Pseudomonas aeruginosa* ATCC 9027, *Staphylococcus aureus* ATCC 29213, *Candida albicans* ATCC 2091, *Enterococcus hirae* ATCC 10541, *Proteus vulgaris* ATCC 6380, and *Staphylococcus epidermidis* ATCC 12228. (**A**–**A^2^**)—BSS11, (**B**–**B^2^**)—BSS17, (**C**–**C^2^**)—BSS19.

**Figure 2 molecules-28-03210-f002:**
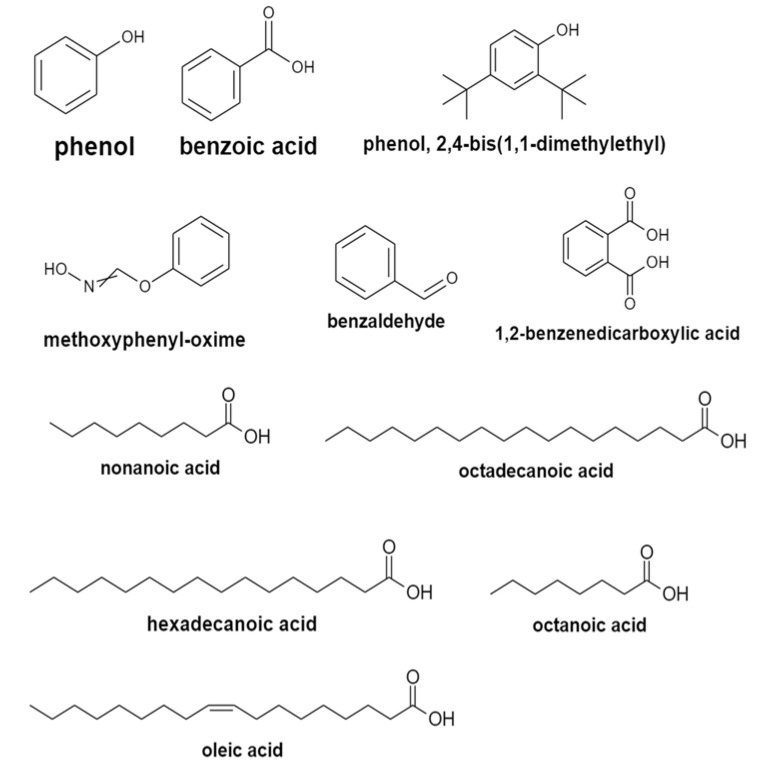
Structure of the components identified from *Bacillus* spp. isolates.

**Figure 3 molecules-28-03210-f003:**
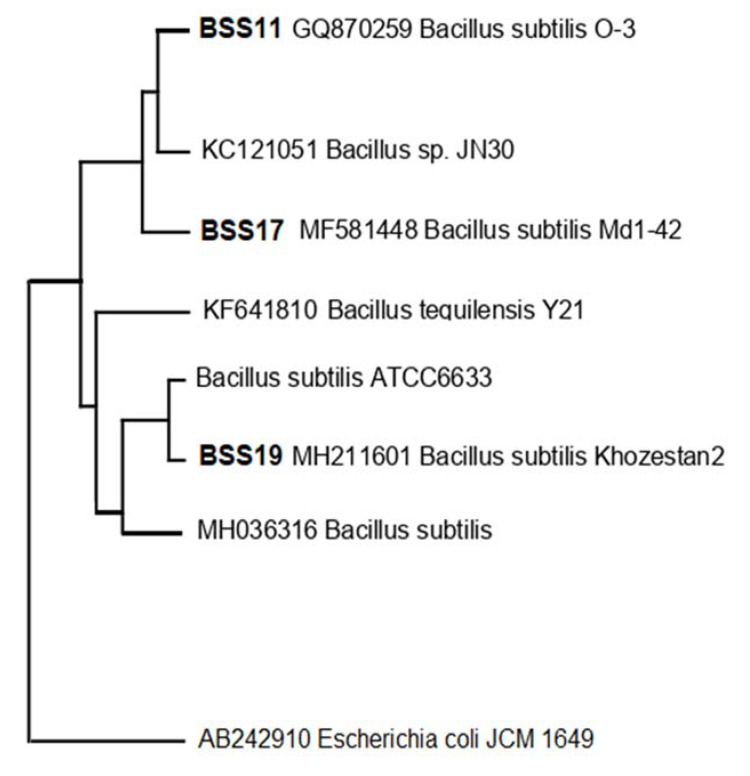
The phylogenetic tree using the neighbor-joining model was constructed based on 16S rRNA gene sequences representing different *Bacillus subtilis* subspecies, i.e., *Bacillus subtilis* O-3, *Bacillus subtilis* Md1-42, and *Bacillus subtilis* Khozestan2, respectively. As an outgroup, *E. coli JCM 1649 (AB242910)* was used.

**Table 1 molecules-28-03210-t001:** Colony morphology and microscopic presentation of the isolated bacterial species.

Bacterial Species	Media	Colony Color and Texture	Microscopic Presentation
*Bacillus subtilis* O-3 (BSS11)	bacillus medium	white, irregular, flat	gram positive, spore-forming, rod.
*Bacillus subtilis* Md1-42 (BSS17)	bacillus medium	white, irregular, flat	gram positive, spore-forming, rod.
*Bacillus subtilis* Khozestan2 (BSS19)	bacillus medium	white, irregular, flat	gram positive, spore-forming, rod.

**Table 2 molecules-28-03210-t002:** Antibacterial activity of the bacterial culture extracts against pathogenic strains.

Species of Microorganism	BSS11 (C), mm	BSS11 (BC), mm	BSS11 (SC), mm	BSS17 (C), mm	BSS17 (BC), mm	BSS17 (SC), mm	BSS19 (C), mm	BSS19 (BC), mm	BSS19 (SC), mm	*Control (Streptomycin)*
*Staphylococcus aureus* ATCC 29213	19 ± 1.33 *	21 ± 1.33 *	25 ± 1.53 *	22 ± 0.50 *	26 ± 0.44 *	30 ± 2.50 *	22 ± 1.55	26 ± 1.24	29 ± 1.54	29 ± 0.33 ***
*Staphylococcus epidermidis* ATCC 12228	18 ± 1.20 *	20 ± 1.33 *	25 ± 1.20 *	19 ± 1.00	20 ± 1.26	22 ± 1.50	18 ± 1.42 *	18 ± 1.44 *	19 ± 1.54 *	22 ± 0.33 ***
*Streptococcus group B*	16 ± 1.20	15 ± 1.00	19 ± 1.20	13 ± 1.16 *	15 ± 1.14 *	17 ± 1.15 *	15 ± 1.33 *	14 ± 1.17 *	16 ± 1.33 *	22 ± 0.33 ***
*Streptococcus mutans* ATCC 25175	16 ± 1.22 *	17 ± 1.22 *	17 ± 1.82 *	14 ± 1.33	16 ± 1.33	19 ± 1.33	18 ± 0.31 *	19 ± 1.00 *	20 ± 1.32 *	16 ± 0.33 ***
*Candida albicans* ATCC 2091	27 ± 2.00 *	25 ± 1.50 *	30 ± 2.50 *	29 ± 1.00	31 ± 1.22	35 ± 1.26	38 ± 1.49	35 ± 1.62	40 ± 1.22	39 ± 0.33 ***
*Candida krusei* ATCC 14243	25 ± 2.33 *	27 ± 2.33 *	29 ± 2.35 *	23 ± 1.38	25 ± 0.33	25 ± 1.34	23 ± 2.00 *	26 ± 1.66 *	27 ± 2.00 *	35 ± 0.33 ***
*Pseudomonas aeruginosa* ATCC 9027	14 ± 1.22	17 ± 0.33	17 ± 1.51	13 ± 0.44	12 ± 1.44	13 ± 1.10	20 ± 1.27 *	20 ± 1.53 *	22 ± 1.81 *	23 ***
*Shigella sonnei* ATCC 25931	0	0	0	0	0	0	0	0	0	19 ± 0.33 ***
*Klebsiella pneumonia* ATCC 13883	0	0	0	0	0	0	0	0	0	15 ± 0.33 ***
*Salmonella enterica* ATCC 35664	0	0	0	0	0	0	0	0	0	19 ± 0.33 ***
*Klebsiella aerogenes* ATCC 13048	14 ± 1.82 *	17 ± 1.57 *	17 ± 1.82 *	9 ± 1.33 *	12 ± 1.22 *	13 ± 1.63 *	16 ± 0.53 *	13 ± 1.53 *	16 ± 1.53 *	17 ± 0.33 ***
*Enterococcus hirae* ATCC 10541	0	0	0	0	0	0	0	0	0	20 ± 0.33 ***
*Escherichia coli* ATCC 25922	18 ± 1.22 *	19 ± 1.57 *	20 ± 1.24 *	22 ± 1.44 *	20 ± 1.31 *	22 ± 1.54 *	19 ± 0.49	22 ± 0.46	22 ± 1.17	23 ± 0.33 ***
*Serratia marcescens* ATCC 13880	13 ± 1.22 *	15 ± 0.33 *	15 ± 1.56 *	17 ± 1.22	18 ± 1.55	18 ± 1.64	0	0	0	24 ± 0.33 ***
*Proteus Vulgaris* ATCC 6380	17 ± 1.33 *	16 ± 0.33 *	18 ± 1.34 *	11 ± 1.33 *	11 ± 1.54 *	13 ± 1.53 *	0	0	0	22 ± 0.33 ***

* Data are represented as the means ± SE (n = 3). Values with same superscript symbols are not statistically different. Significance level * < ***.

**Table 3 molecules-28-03210-t003:** Antibiotic resistance profile of the *Bacillus* strains.

Antibiotic (AB, Charge in μg) Used		*Bacillus* Strains					
		BSS11		BSS17		BSS19	
		Diameter (mm)	S/R	Diameter (mm)	S/R	Diameter (mm)	S/R
**Penicillins**	Penicillin G (PEN, 10)	30 ± 0.98 ^bc^	**S**	24 ± 0.56 ^ab^	**S**	23 ± 0.29 ^a^	**S**
	Ampicillin (AMP, 10)	30 ± 1.43 ^ab^	**S**	27 ± 1.34 ^ab^	**S**	27± 0.38 ^a^	**S**
	Amoxycillin (AMOX, 30)	32 ± 0.98 ^abc^	**S**	30 ± 1.30 ^abc^	**S**	12 ± 0.29 ^ab^	**S**
	Amoxycillin-clavulanic acid (AMC, 30)	28 ± 1.05 ^bcd^	**S**	23 ± 0.33 ^abc^	**S**	0 ± 0.00 ^b^	**S**
	Carbenicillin (CAR, 100)	38 ± 0.28 ^abc^	**S**	28 ± 0.35 ^ab^	**S**	10 ± 0.28 ^abc^	**S**
	Cloxacillin (CX, 5)	0 ± 0.00 ^b^	**R**	0 ± 0.00 ^b^	**R**	0 ± 0.00 ^b^	**R**
**Macrolides**	Erythromycin (ERO, 15)	35 ± 0.31 ^ab^	**S**	32 ± 1.41 ^abc^	**S**	30 ± 0.23 ^a^	**S**
	Azithromycin (AZM, 15)	36 ± 0.28 ^a^	**S**	35 ± 1.43 ^ab^	**S**	27 ± 1.33 ^ab^	**S**
**Cephalosporins**	Cefepime (FEP, 30)	35 ± 0.51 ^ab^	**S**	25 ± 0.57 ^a^	**S**	38 ± 0.86 ^ab^	**S**
	Cefepime/clavulanic acid FEC-40	36 ± 0.98 ^a^	**S**	30 ± 0.36 ^a^	**S**	39 ± 0.67 ^a^	**S**
	Cephalatin (KF, 30)	32 ± 0.33 ^ab^	**S**	40 ± 0.37 ^ab^	**S**	26 ± 1.32 ^abc^	**S**
	Cefotaxime (CTX, 30)	26 ± 0.98 ^a^	**S**	40 ± 0.52 ^a^	**S**	24 ± 0.98 ^ab^	**S**
**Aminoglycosides**	Gentamicin (CN, 120)	40 ± 0.28 ^ab^	**S**	40 ± 0.27 ^ab^	**S**	38 ± 0.28 ^ab^	**S**
	Streptomycin (STR, 10)	26 ± 0.19 ^ab^	**S**	25 ± 0.48 ^ab^	**S**	22 ± 0.20 ^ab^	**S**
	Tobramycin (TOB, 10)	33 ± 0.98 ^a^	**S**	39± 1.18 ^a^	**S**	22 ± 0.23 ^abc^	**S**
**Tetracyclines**	Tetracycline (TET, 30)	35 ± 0.28 ^a^	**S**	30 ± 0.33 ^a^	**S**	20 ± 1.18 ^ab^	**S**
**Polypeptides**	Polymyxin (PB, 300)	0 ± 0.00 ^b^	**R**	0 ± 0.00 ^b^	**R**	0 ± 0.00 ^b^	**R**
	Bacitromycin (B, 10)	0 ± 0.00 ^b^	**R**	0 ± 0.00 ^b^	**R**	0 ± 0.00 ^b^	**R**

The Newmann–Keuls test shows that the averages affected by the different superscript letters in the row and column are significantly different at the 5% level. Values are the means ± standard error; legend: D = dimension, S/R = sensible/resistant.

**Table 4 molecules-28-03210-t004:** The main constituents of bacterial extract BSS11 identified through a GC–MS analysis.

*Bacillus subtilis* O-3 (BSS11)							
No	Name	Molecular Formula	Molecular Mass, g/mol	Retention Time (min)	Pubchem Compound CID	Similarities	Area, %
1	2-Butanone	C_4_H_8_O	72.11	2.111	6569	75	9.68622
2	Acetic acid ethenyl ester	C_4_H_6_O_2_	86.09	2.63	7904	63	3.553507
3	2-Pentanone, 3-methyl-	C_6_H_12_O	100.16	2.993	11262	80	3.452664
4	Disulfide, dimethyl	C_2_H_6_S_2_	94.2	3.578	12232	81	2.579829
5	2-Heptanone, 6-methyl-	C_8_H_16_O	128.21	5.602	13572	85	2.012434
6	Pyrazine, methyl-	C_5_H_6_N_2_	94.11	5.95	7976	93	4.364266
7	Pyrazine, 2,5-dimethyl-	C_6_H_8_N_2_	108.14	6.722	31252	92	4.600645
8	1-Hexanol	C_6_H_14_O	102.17	7.055	8103	74	1.171664
9	Pyrazine, trimethyl-	C_7_H_10_N_2_	122.17	7.824	26808	75	1.113828
10	Pyrazine, 3-ethyl-2,5-dimethyl-	C_8_H_12_N_2_	136.19	8.355	25916	80	2.579326
11	1-Hexanol, 2-ethyl-	C_8_H_18_O	130.229	8.864	7720	84	0.5058
12	Pyrrole	C_4_H_5_N	67.09	9.166	8027	90	0.639997
13	Benzaldehyde	C_7_H_6_O	106.12	9.374	240	82	7.155098
14	Acetic acid, trifluoro-, nonyl ester	C_11_H_19_F_3_O_2_	240.26	10.344	6428483	73	0.416699
15	(S)-(+)-6-Methyl-1-octanol	C_9_H_20_O	144.25	10.605	13548104	79	0.560066
16	Oxime-, methoxy-phenyl-_	C_8_H_9_NO_2_	151.16	12.025	9602988	70	2.008104
17	Acetamide	C_2_H_5_NO	59.07	12.1	178	96	11.58042
18	Propanamide	C_3_H_7_NO	73.09	12.612	6578	71	0.730287
19	2,4-Decadienal, (E,E)-	C_10_H_16_O	152.23	12.825	5283349	68	0.893933
20	Hexanoic acid	C_6_H_12_O_2_	116.16	13.043	8892	60	0.301542
21	2-Tetradecanone	C_14_H_28_O	212.37	13.441	75364	86	1.671165
22	(R)-(−)-4-Methylhexanoic acid	C_7_H_14_O_2_	130.18	13.913	12600623	70	0.496031
23	Hexanoic acid, 2-ethyl-	C_8_H_16_O_2_	144.21	14.172	8697	66	0.587733
24	Phenol	C_6_H_6_O	94.11	14.738	996	96	4.246424
25	Octanoic acid	C_8_H_16_O_2_	144.21	15.283	379	67	0.803864
26	2,4,7,9-Tetramethyl-5-decyn-4,7-diol	C_14_H_26_O_2_	226.35	15.658	31362	69	0.481144
27	Nonanoic acid	C_9_H_18_O_2_	158.24	16.331	8158	82	1.003712
28	2-Octyl benzoate	C_15_H_22_O_2_	234.33	17.33	243800	66	0.967435
29	Phenol, 2,4-bis(1,1-dimethylethyl)-	C_14_H_22_O	206.32	17.784	7311	87	1.242429
30	Benzoic acid, pentyl ester	C_12_H_16_O_2_	192.25	18.345	16296	68	0.712256
31	Benzoic acid	C_7_H_6_O_2_	122.12	18.705	243	85	0.94121
32	1,2-Benzenedicarboxylic acid, bis(2-methylpropyl) ester	C_16_H_22_O_4_	278.34	19.822	6782	93	2.287847
33	Dibutyl phthalate	C_16_H_22_O_4_	278.34	21.072	3026	74	2.75918
34	Hexadecanoic acid	C_16_H_32_O_2_	256.42	22.573	985	71	6.166636
35	Oleic Acid	C_18_H_34_O_2_	282.5	24.497	445639	82	8.522779
36	9,12-Octadecadienoic acid (Z,Z)-	C_18_H_32_O_2_	280.4	25.011	5280450	71	3.023523

**Table 5 molecules-28-03210-t005:** The main constituents of bacterial extract BSS17 identified through a GC–MS analysis.

*Bacillus subtilis* Md1-42 (BSS17)							
No	Name	Molecular Formula	Molecular Mass, g/mol	Retention Time (min)	Pubchem Compound CID	Similarities	Area, %
1	(2-Aziridinylethyl)amine	C_4_H_10_N_2_	86.14	1.157	97697	78	0.440368
2	1-Propen-2-ol, acetate	C_5_H_8_O_2_	100.12	1.664	7916	76	0.91485
3	2,3-Butanedione	C_4_H_6_O_2_	86.09	2.649	650	93	17.04656
4	3-Penten-1-ol	C_5_H_10_O	86.13	5.699	510370	83	0.269452
5	Acetoin	C_4_H_8_O_2_	88.11	6.247	179	76	37.17943
6	3-Pentanol, 2-methyl-	C_6_H_14_O	102.17	7.009	11264	80	1.154491
7	2-Nonen-1-ol	C_9_H_18_O	142.24	7.108	61896	68	0.420964
8	2-Hydroxy-3-pentanone	C_5_H_10_O_2_	102.13	7.215	521790	81	1.137636
9	Ethane-1,1-diol dibutanoate	C_10_H_18_O_4_	202.25	8.244	551339	77	0.624527
10	Acetic acid	C_2_H_4_O_2_	60.05	8.355	176	93	0.854423
11	1-Hexanol, 2-ethyl-	C_8_H_18_O	130.229	8.889	7720	84	0.206645
12	Benzaldehyde	C_7_H_6_O	106.12	9.399	240	92	2.022607
13	2,3-Butanediol	C_4_H_10_O_2_	90.12	9.483	262	78	3.407863
14	1,6-Octadien-3-ol, 3,7-dimethyl-	C_10_H_18_O	154.25	9.632	6549	83	0.530564
15	Propanoic acid, 2-methyl-	C_4_H_8_O_2_	88.11	9.833	6590	82	3.007214
16	2,3-Butanediol, [R-(R*,R*)]-	C_4_H_10_O_2_	90.12	9.927	225936	75	0.516676
17	1-Nonanol	C_9_H_20_O	144.25	10.367	8914	79	0.235951
18	(S)-(+)-6-Methyl-1-octanol	C_9_H_20_O	144.25	10.63	13548104	86	0.668355
19	Butanoic acid, 2-methyl-	C_5_H_10_O_2_	102.13	11.076	8314	81	2.487502
20	Oxime-, methoxy-phenyl-_	C_8_H_9_NO_2_	151.16	12.051	151.16	70	0.59943
21	2,4-Decadienal	C_10_H_16_O	152.23	12.855	5283349	70	0.310428
22	2,2,4-Trimethyl-1,3-pentanediol diisobutyrate	C_16_H_30_O_4_	286.41	13.615	23284	73	0.259067
23	(R)-(−)-4-Methylhexanoic acid	C_7_H_14_O_2_	130.18	13.942	12600623	74	0.173808
24	Phenol	C_6_H_6_O	94.11	14.772	996	94	0.427543
25	Octanoic acid	C_8_H_16_O_2_	144.21	15.313	379	68	0.235055
26	Nonanoic acid	C_9_H_18_O_2_	158.24	16.358	8158	87	0.268447
27	Hexadecanoic acid, methyl ester	C_17_H_34_O_2_	270.5	17.011	8181	89	0.296456
28	2-Octyl benzoate	C_15_H_22_O_2_	234.33	17.353	243800	67	0.301277
29	Benzoic acid, heptyl ester	C_14_H_20_O_2_	220.31	18.076	81591	75	0.189386
30	Benzoic acid, undecyl ester	C_18_H_28_O_2_	276.4	18.368	229159	70	0.218367
31	Benzoic acid	C_7_H_6_O_2_	122.12	18.538	243	84	0.270263
32	1,2-Benzenedicarboxylic acid, bis(2-methylpropyl) ester	C_16_H_22_O_4_	C_16_H_22_O_4_	19.238	6782	91	0.666223
33	Phenol, 2,4-bis(1,1-dimethylethyl)-	C_14_H_22_O	206.32	19.849	7311	86	0.746427
34	Oleic Acid	C_18_H_34_O_2_	282.5	20.98	445639	68	0.295063
35	Dibutyl phthalate	C_16_H_22_O_4_	278.34	21.113	3026	82	1.029155
36	Hexadecanoic acid	C_16_H_32_O_2_	256.42	22.592	985	85	3.51103
37	Octadecanoic acid	C_18_H_36_O_2_	284.5	24.242	5281	68	3.107501
38	Oleic Acid	C_18_H_34_O_2_	282.5	24.525	445639	84	5.557709
39	9,12-Octadecadienoic acid (Z,Z)-	C_18_H_32_O_2_	280.4	25.047	5280450	84	9.157715

**Table 6 molecules-28-03210-t006:** The main constituents of bacterial extract BSS19 identified through a GC–MS analysis.

*Bacillus subtilis* Khozestan2 (BSS19)							
No	Name	Molecular Formula	Molecular Mass, g/mol	Retention Time (min)	Pubchem Compound CID	Similarities	Area, %
1	Carbamic acid, monoammonium salt	CH_6_N_2_O_2_	78.071	1.134	517232	88	1.700997
2	Ethanol	C_2_H_6_O	46.07	2.234	702	81	0.46569
3	1-Butanol	C_4_H_10_O	74.12	4.321	263	91	6.044015
4	2-Heptanone, 6-methyl-	C_8_H_16_O	128.21	5.599	13572	85	1.508914
5	2-Heptanone, 5-methyl-	C_8_H_16_O	128.21	5.839	28965	87	0.899369
6	Pyrazine, 2,5-dimethyl-	C_6_H_8_N_2_	108.14	6.717	31252	94	6.575729
7	Acetic acid	C_2_H_4_O_2_	60.05	8.318	176	93	4.27265
8	2-Decanone	C_10_H_20_O	156.26	8.399	12741	76	3.555438
9	1-Hexanol, 2-ethyl-	C_8_H_18_O	130.229	8.859	7720	85	1.019178
10	Benzaldehyde	C_7_H_6_O	106.12	9.361	240	91	3.066211
11	1-Octene, 6-methyl-	C_9_H_18_	126.24	9.587	518716	77	0.767838
12	Propanoic acid, 2-methyl-	C_4_H_8_O_2_	88.11	9.796	6590	85	8.705907
13	1-Octanol, 2-methyl-	C_9_H_20_O	144.25	9.896	102495	82	0.837655
14	1-Nonanol	C_9_H_20_O	144.25	10.342	8914	81	1.748035
15	(S)-(+)-6-Methyl-1-octanol	C_9_H_20_O	144.25	10.604	13548104	90	3.260679
16	Benzeneacetaldehyde	C_8_H_8_O	120.15	10.821	998	69	2.227017
17	2-Furanmethanol	C_5_H_6_O_2_	98.1	10.918	7361	87	1.948174
18	Hexanoic acid, 2-methyl-	C_7_H_14_O_2_	130.18	11.036	20653	80	25.50356
19	2-Dodecanone	C_12_H_24_O	184.32	11.198	22556	77	1.361537
20	Oxime-, methoxy-phenyl-_	C_8_H_9_NO_2_	151.16	12.018	9602988	65	1.54511
21	2(5H)-Furanone	C_4_H_4_O_2_	84.07	12.114	10341	77	2.40381
22	Formamide	CH_3_NO	45.041	12.32	713	77	0.972164
23	2-Tetradecanone	C_14_H_28_O	212.37	13.44	75364	80	1.097307
24	Maltol	C_6_H_6_O_3_	126.11	14.375	8369	93	4.442068
25	Ethanone, 1-(1H-pyrrol-2-yl)-	C_6_H_7_NO	109.13	14.429	14079	67	1.456046
26	Phenol	C_6_H_6_O	94.11	14.739	996	96	6.642077
27	4H-Pyran-4-one, 2,3-dihydro-3,5-dihydroxy-6-methyl-	C_6_H_8_O_4_	144.12	17.293	119838	88	1.650456
28	Benzoic acid, hept-2-yl ester	C_14_H_20_O_2_	220.31	18.053	243678	75	0.421796
29	Benzoic acid	C_7_H_6_O_2_	122.12	18.698	243	91	1.317335
30	5-Hydroxymethyldihydrofuran-2-one	C_5_H_8_O_3_	116.11	19.196	98431	73	0.384416
31	1,2-Benzenedicarboxylic acid, bis(2-methylpropyl) ester	C_16_H_22_O_4_	C_16_H_22_O_4_	19.356	6782	91	0.986222
32	Phenol, 2,4-bis(1,1-dimethylethyl)-	C_14_H_22_O	206.32	19.786	7311	85	1.085735

**Table 7 molecules-28-03210-t007:** Identification of the bacterial species based on the sequence similarities.

No	Isolates	16S rRNA Amplified Region Length	% Similarity	NCBI Accession No
1	BSS11	1443 bp	99% with *Bacillus subtilis* O-3	GQ870259
2	BSS17	1454 bp	99% with *Bacillus subtilis* Md1-42	MF581448
3	BSS19	1450 bp	99% with *Bacillus subtilis* Khozestan2	MH036316

## Data Availability

The authors confirm that the data supporting the findings of this study are available within the article.

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
