# Peer review of "Molecular Characterization of Some Bacillus Species from Vegetables and Evaluation of Their Antimicrobial and Antibiotic Potency"

_molecules, 2023, doi:10.3390/molecules28073210_

Round 1

Reviewer 1 Report

The research article entitled “Molecular Characterization of some Bacillus species from vegetables and Evaluation of their Antimicrobial and Antibiotic Potencyis written well and could be accepted in “Molecules” after some minor corrections.

1. Abstract: Add at least two new keywords. The keywords should be arranged alphabetically in the manuscript.

2. Abstract should be shortened at least to 250 words.

3. Line 50. Reference citation format should be written as; [2-5].

4. Line 286. Reference citation format should be written as; [24-27].

5. How the authors measure the zones by electronic ruler?

6. Does the vegetable type have any role in antibiotic potency?

7. The use of innovative antimicrobial compounds to separate bacteria from the soil has also been described in several other key types of study. Revise this sentence.

8. How the extraction was done. Which solvent was used? Please mention in manuscript?

9. Normally extraction is done in organic solvent. Carbon dioxide is inorganic. From where carbon dioxide came in GCMS results?

10.Which reported compound has in GCMS results has highest antibiotic potential?

11. What is the practical potential and application of this research work.

12. Normally bacteria are pathogenic. If we use medicinal plant based plant extract as alternative for antimicrobial purposes it will be better; then what is advantage of your work over plant based extracts 

13. There is no description about authors’ contribution?

14. Check the plagiarism of the manuscript.

15. The manuscript should be revised carefully. There are some minor grammatical mistakes in manuscript.

Reviewer 2 Report

This manuscript needs careful revision. At this point, the manuscript is a collection of unrelated results that have not been properly analyzed. At the same time, the data itself is of undoubted interest.

Introduction section.

The introduction does not make it clear to the reader why the authors carried out this study. The text of the section is a disorderly alternation of literature data on the potential danger of microorganisms with resistance to antibiotics, the need to make new antibiotics, and the biological properties of both the antibiotics themselves and their producers. The purpose of the study is not clearly formulated.

The reviewer believes that the introduction should inform readers about the bacteria of the genus Bacillus themselves and the range of substances produced by them that can and do have antimicrobial activity; especially since there have already been many such works. On the other hand, bacteria of the genus Bacillus, including those synthesizing antibiotic-like substances, are a common cause of food spoilage, mainly due to the production of a large number of extracellular hydrolases, which can inhibit other microorganisms. That is, the introduction section can be significantly improved.

Section of materials and methods.

2.1. Isolation of potential strains of the genus Bacillus spp.

This should state what Nutrient Agar (NA) is and why it was used. There is NB in line 92, perhaps this is a mistake, but the authors meant NA.

Line 105 contains the term "endophyte", which is misleading as it refers specifically to Bacillus spp.

In lines 109-113 sentences are duplicated. Here, the meaning of the phrase "The cultures were then eliminated using three different techniques" is not clear. How are the cells removed? Or something different?

Line 116, is it legal to use the term "organic supernatant"? Apparently, here we are talking about ethyl acetate, which contains extracted substances?

Here; the authors write about the study of the antibacterial activity of the extract by the method of well diffusion. How were the concentrations of substances in the studied extracts standardized? The diameter of the zone of inhibition will be determined to some extent by the concentration of antimicrobial substances.

Lines 162-164. It's not clear here. Do you mean the relationship between the antimicrobial activity of extracts and the concentration of certain substances identified in these extracts by gas chromatography?

And yes, why was the sensitivity to antibiotics determined in bacilli?

Results section.

From the photographs shown in Figure 1 it is difficult to understand the growth of which microorganisms are inhibited by the studied strains of bacilli.

In table 2, line 216 is duplicated by BSS17, here, apparently, one of the strains should be BSS19.

Data on the antimicrobial activity of which extract are presented in this table? And where are the data on other extracts? The materials and methods indicate that three extracts were obtained, and their antimicrobial activity was evaluated.

Again, the question arises about the appropriateness of determining the sensitivity to antibiotics in the studied bacilli (table 3, line 229).

Lines 234-250, subsection 3.5. Here are the results of gas chromatography in the form of three large and difficult to understand tables. Did the strains differ in chemical composition? Are the differences in chemical composition somehow related to antimicrobial activity? There are no answers to these questions here because there is no analysis of the results obtained by the authors. And yet, many substances with antimicrobial activity (those produced by bacilli) cannot be identified using gas chromatography, why was only this method used?

Discussion section.

This section is bad because it looks like a review of the literature on this issue; here there are inlays from the results obtained by the authors. But there is no analysis of the authors' own results here, so the discussion section is effectively absent.

The conclusions are declarative and are not based on the results obtained by the authors.